# ParaScopes: What do Language Models Activations Encode About Future Text?

## Abstract

Interpretability studies in language models often investigate forward-looking representations of activations. However, as language models become capable of doing ever longer time horizon tasks, methods for understanding activations often remain limited to testing specific concepts or tokens. We develop a framework of Residual Stream Decoders as a method of probing model activations for paragraph-scale and document-scale plans. We test several methods and find information can be decoded equivalent to 5+ tokens of future context in small models. These results lay the groundwork for better monitoring of language models and better understanding how they might encode longer-term planning information.

## 1 The Planning Decodability Hypothesis

Large Language Models (LLMs) generate coherent multi-paragraph text through autoregressive prediction. However, coherence over increasingly long time horizons (Kwa et al., 2025) suggests some degree of forward-thinking in writing. In this paper, rather than asking whether models "plan" in an anthropomorphic sense, we operationalize a test for planning: planning at scale $X$ exists if information about scale-$X$ content is decodable from activations prior to being generated. This planning **Decodability Hypothesis** makes planning empirically testable while remaining agnostic about the underlying mechanisms, yet presents two main potential issues. The first concern is that a sufficiently complex probe may implicitly infer plans that are not actually present due to correlations. The second is that negative results would not necessarily rule out the existence of plans that our methods cannot decode.

We primarily investigate this hypothesis at the paragraph scale outputs, but also briefly investigate outline of full outputs. We choose this scale because some minimal level of planning is likely to be functionally useful, as maintaining topic coherence requires some representation of upcoming content. Additionally, paragraph boundaries ("\n\n" tokens) provide natural intervention points for studying information transitions. Our investigation reveals that some information is decodable with relative ease in models as small as Llama 3.2 3B, providing evidence of limited planning.

## 2 Related Work

Much interpretability work has been focused on understanding the hidden-layer activations of language models. There are various methods ranging from simple Logit Lens analysis (nostalgebraist, 2020), to circuit analysis (Elhage et al., 2021; Olsson et al., 2022; Wang et al., 2022), to concept analysis with Sparse Auto Encoders Cunningham et al. (2023); Anthropic Interpretability Team (2023); Gao et al. (2024). Additionally, there is much work on probing models (Belinkov, 2022), predominantly for a single trait Burns et al. (2023) or specific aspect(s) (Tenney et al., 2019; Zou et al., 2023) of the output. Each of these predominantly focuses on understanding the effect of the layer activations on some immediate token or concept.

Recent work has investigated forward-thinking mechanisms in language models (Vaswani et al., 2017) across multiple scales and methodological approaches. At the token level, Pal et al. (2023) pioneered probing hidden states to predict future tokens, while Wu et al. (2024) distinguished between "pre-caching" versus "breadcrumbs" explanations for future-oriented representations. Moving to higher semantic levels, Pochinkov et al. (2024) and Ghandeharioun et al. (2024) introduced limited

methods for decoding model activations by transferring them to a new context, and we build upon this work.

Mechanistic circuit analysis has provided some of the strongest evidence for planning. Lindsey et al. (2025) demonstrated "Planning in Poems" through circuit tracing, while their companion work on "Hidden Goals" (Marks et al., 2025) showed that models can represent goal states that guide generation. In specialized domains, Jenner et al. (2024) found evidence of learned look-ahead in chess-playing networks, and Taufeeque et al. (2024) studied planning in block movement games. This paper aims to provide a more general method for studying longer-horizon planning in LLMs.

The relationship between agency and planning has also received attention, with Li et al. (2024) finding that RLHF changes the way models represent future information, leading to more structured generation patterns. Methodologically, work on representation engineering (Zou et al., 2023), activation patching (Turner et al., 2023), and interpretability frameworks like Patchscopes (Ghandeharioun et al., 2024) and LatentQA (Pan et al., 2024), have all developed tools applicable to planning research, without directly focusing on interpretability of longer-term planning.

Additionally, there is a variety of work that tries to modify model training to explicitly plan ahead, either to improve inference speed (Bhendawade et al., 2024), or to try to get the model to write explicit plans (LCM Team, 2024; Yin et al., 2024). The focus of this paper is instead on understanding how existing transformers might already encode future text information.

## 3  RESIDUAL STREAM DECODERS AND PARASCOPES

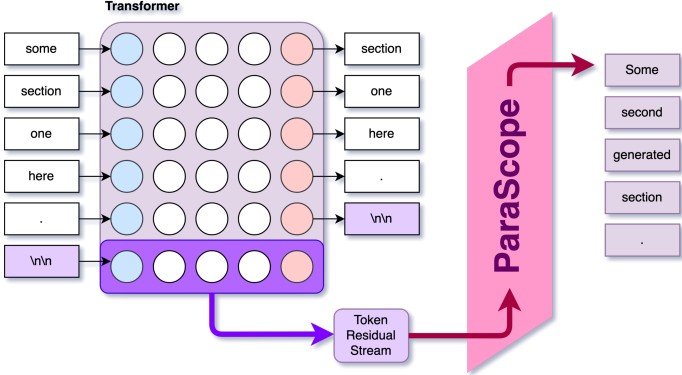

Figure 1: Simple diagram showing the idea behind ParaScopes. The residual stream of an LLM is taken at a specific point, and we try to use ParaScope methods to infer what the LLM might say next.

When language models generate text, they likely need to maintain some form of forward thinking to produce coherent output. While there are many aspects of cognition that one can call "planning", we focus on a specific operationalizable question: Does a language model encode information about its likely future outputs within its internal representations?

To investigate planning, we need a systematic way to extract information from a model's internal representations. We define a **Residual Stream Decoder** (RSD) as any method that can reconstruct future content from current activations better than chance.

Formally, given a transformer's residual stream $R_i \in \mathbb{R}^{L \times d}$ at position $i$, a decoder consists of:

- An activation extractor that selects relevant information from $R_i$
- A mapping function that transforms this information into predictions about future content
- An evaluation metric that measures how well these predictions match actual outcomes

The key insight is that if models truly maintain forward-looking information, then decoders trained on this internal state should outperform baselines that lack access to the model's "planning" representations. This information-theoretic perspective allows us to sidestep debates about whether models

truly "plan" in a cognitive sense, instead asking what forward-looking information can be decoded from the model's activations, and use these as methods to test the Decodability Hypothesis.

We additionally coin the term **ParaScope** to be a Residual Stream Decoder specifically used to decode the next paragraph or section of text. See Figure 1 for a simple illustration. for a basic illsimple illustration.

### 3.1 CORE METHODS AND SETUP

We examine how language models encode information about upcoming paragraphs within their residual stream. Our focus is on the transition points marked by "\n\n," where we hypothesize that the model may already represent aspects of the next section. Throughout this work, we treat each span of text separated by this delimiter as a paragraph. In practice, this boundary aligns well with the model's own segmentation and supports paragraph-level analysis of planning.

For our experiments, we use Llama-3.2-3B-Instruct (Dubey et al., 2024) as the primary model, with temperature=0.3 for all generation tasks. We additionally employ SONAR models (Duquenne et al., 2023) for text auto-encoding (TAE) tasks, and Qwen 3 embedding model (Zhang et al., 2025) for text vectorization as a method to compare outputs.

For our dataset, we require data that represents the outputs of the LLM so we can understand its thinking process. We use synthetic question prompts based on FineWeb-Edu (Lozhkov et al., 2024), and generate a diverse set of LLM model outputs based on each prompt, up to 1 million text examples with a temperature of 0.3. See Appendix A for more details.

In order to evaluate how well our extraction of potential future context worked, we also need a way to compare similarity between text outputs. The main methods we use are: 1. traditional methods including cosine similarity from text-embed models (Zhang et al., 2025), and BLEURT-20 (Pu et al., 2021), and 2. LLM-as-a-Judge (Zheng et al., 2023b; Liu et al., 2023; Kim et al., 2024b; or Various, 2025) with rubric scoring. For our LLM rubric, we focus on relevant topics in text, coherence, subject match, entity preservation, and detail preservation. For more details on the rubric and our setup, see Appendix C

#### 3.1.1 RESIDUAL STREAM DECODER METHODS

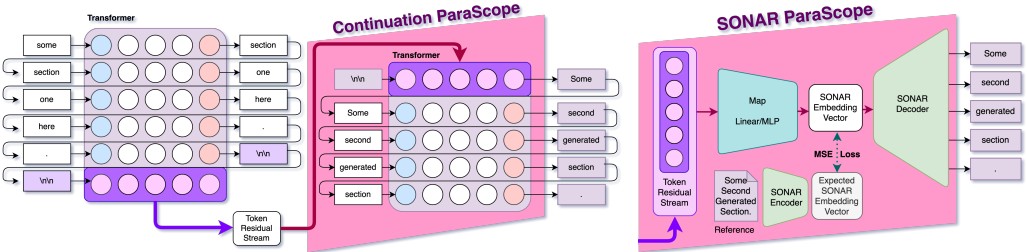

Figure 2: Continuation Parascope (Left) and TAE Parascope (Right). The former takes the whole residual stream of the model and passes it into a blank-context copy of the model for decoding. The latter takes the residual stream of the model and trains a map to output a text autoencoder vector.

We introduce two complementary Residual Stream Decoder approaches (as shown in Figure 2):

**Continuation ParaScope:** This method, inspired by Ghandeharioun et al. (2024) and Pochinkov et al. (2024), intervenes directly on model activations to probe what the model might generate. We insert a blank prompt consisting of '<bos>\n\n', replace the residual stream activations of the '\n\n' token with the activations saved from an original generation, then generate up to 128 tokens. This approach requires no training, instead relying on the model to access any information it may have stored.

**TAE ParaScope:** the Text AutoEncoder ParaScope learns a mapping from the residual stream to a structured text auto-encoder embedding space from the SONAR model. We normalize the residual

stream activations (treating each dimension as independent, using mean and standard deviation normalization) and train a map to predict the TAE embedding vectors of the upcoming paragraph.

We choose to use a linear map, which takes the normalized residual diffs from some subset of layers, and outputs a SONAR vector of dimension 1024. See Appendix H.1 for details.

The predicted embedding vectors are then decoded using SONAR's decoder to produce human-readable text. This structured approach allows us to leverage the semantic understanding encoded in the SONAR embedding space.

### 3.1.2 BASE EVALUATION FRAMEWORK

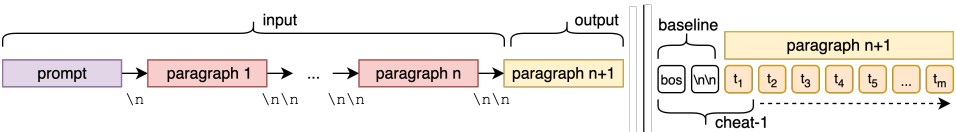

Figure 3: Basic diagram explaining the next-paragraph prediction task (left) and showing how we produce the baseline generation and cheat-k predictions (right)

To evaluate our methods, we must first establish appropriate baselines for comparison.

**Random/Blind Baseline:** Generation with only a blank '<bos>\n\n' context, providing a lower bound worst-case for performance. (See Figure 3)

**Cheat-K Baseline:** Generation with '<bos>\n\n' plus K tokens of the actual upcoming section revealed ($K \in 1, 5, 10$). In Appendix B we briefly investigate the degree to which a model is able to reproduce a similar text from revealing cheat-K tokens, showing saturation at around 20 tokens.

**Regeneration:** Taking the whole previous context (prompt + generated sections up until the current section) and generating what could come next, giving us the ground truth we aim to match.

**Auto-Decoded:** Taking the reference text, encoding it with SONAR, and decoding it again, providing an upper bound on SONAR-based methods.

Additionally, we filter examples shorter than 20 tokens so the "cheat" methods do not have completed examples for fairness (see Appendix D.1 for unfiltered plots).

## 4 PARAGRAPH-LEVEL PLANNING: EVIDENCE AND STRUCTURE

We first test using traditional methods, including cosine similarity from Qwen 3 Embed 0.6B, and BLEURT score. Figure 4 show similar results for both, with standard errors are <0.002. For cosine similarity, we find that Continuation ParaScope (mean 0.39) and TAE ParaScope (0.53) both perform significantly better than random baselines (0.15), while being significantly worse than ground truth (regenerated 0.81, auto-decoded 0.92). Both methods are comparable to the cheat-5 baseline (0.48).

These suggests that the model is storing some limited plans about the future within its activations, giving evidence for the Decodability Hypothesis, comparable to revealing around five future tokens.

We test for more fine-grained features using rubric-based LLM-as-a-judge approach, shown in Figure 5. We primarily compare how well general subjects (on a scale from -1 to 4) and fine-grained details (on a scale from -1 to 3) match the baseline generation (see Appendix E for details).

When looking at subject match, we find that TAE ParaScopes generally seem to outperform Continuation ParaScopes. With a score threshold of $\geq 2$ corresponding to being in a related general domain or better, 75% of TAE ParaScopes achieved this level, compared to 43% for Continuation ParaScopes and 50% for the cheat-10 baseline. This significantly outperforms the random baseline of 0.1%, while still being short of the 99% achieved by ground-truth regeneration.

However, when looking at details match, we find an interesting difference. When it comes to achieving at least Minimal Depth Details (score $\geq 1$, Basic shared details without specifics), both TAE (58%) and Continuation (53%) ParaScopes perform similarly well to each other, and similar to a cheat-5

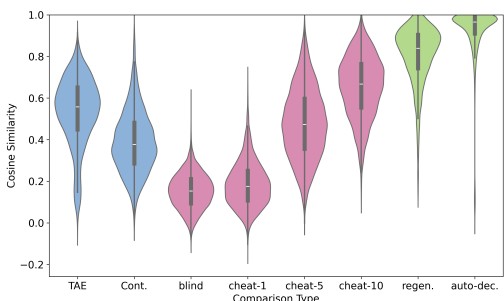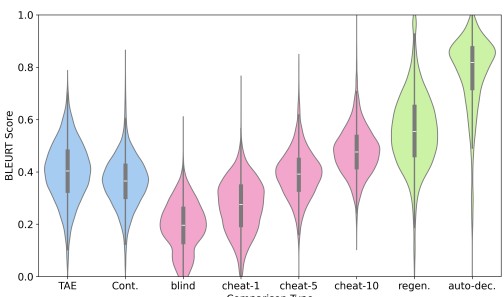

Figure 4: Violin plots showing the performance of TAE ParaScope and Continuation ParaScope against the baselines (0, 1, 5, and 10 tokens) and ground truth (regenerated, auto-decoded) on Cosine sim (left) and BLEURT (right). We filter out short examples < 20 tokens (see Appendix D.1)
.

baseline (55%). However, when it comes to achieving at least Moderate Depth details (score $\geq 2$, Shared details with some supporting facts), Continuation ParaScope (16%) is significantly better than the TAE ParaScope (3%), which in turn falls just short of a cheat-5 baseline (18%). Even ground truth regenerated outputs often fell short, with only 80% achieving a score $\geq 1$ and 18% achieving a score $\geq 2$.

This is qualitatively in line with results from manually looking at examples (see Appendix J), where Continuation ParaScope often either completely miss the task, or give a quite accurate reconstruction.

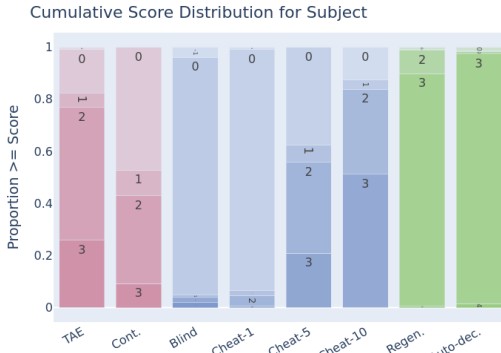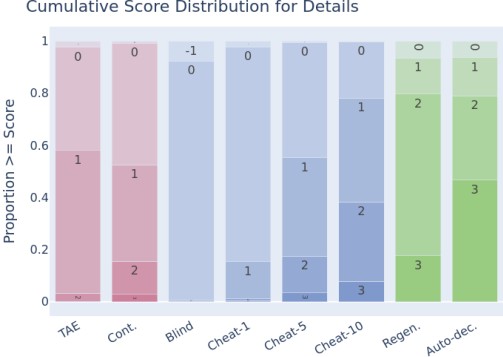

Figure 5: Cumulative bars showing the performance of TAE ParaScope and Continuation ParaScope against the baselines (0, 1, 5, and 10 cheat tokens) and ground truth (regenerated, auto-decoded). (left) shows subject match to original paragraph on a scale from -1 to 4, and (right) shows detail preservation on a scale from -1 to 3. Proportions have standard error of $\pm 0.001$ to $\pm 0.006$

Importantly, both ParaScope methods show significant performance above random baselines, often being able to reconstruct key details about the upcoming text. TAE ParaScope seems to preserve general subject match well, and mostly gets approximate details correct, while Continuation ParaScope seems to work inconsistently but can sometimes restore more fine-grained details. These results give positive evidence for the Decodability Hypothesis, while falling short of ground-truth baselines.

## 5 OUTLINE-LEVEL RESIDUAL STREAM DECODERS

We extend the Decodability Hypothesis to outline-level representations. We take the method of SONAR ParaScopes to construct a TAE Outline RSD, which aims to map residual at the start of generation to a general outline of the text the model is likely to produce. See Figure 6.

We construct a dataset of outlines for this experiment. We take the dataset of model generations we had from Section 3.1, and use Llama 3.2 70B to summarize the text into a bullet-point outline of key

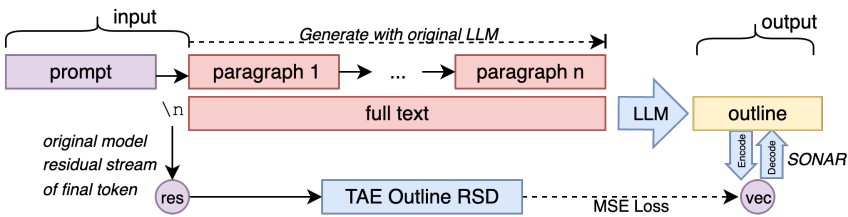

Figure 6: Diagram of experiments for TAE Outlines RSD. We create an outline of the generated dataset, and encode this outline with SONAR. We then train a linear map between the residual stream diffs of the model to the SONAR TAE embedding vector.

details. We then encode these outlines into a SONAR TAE vector to be used as the training target. As the training input, we use the model residuals at the newline token after the prompt.

The architecture for TAE Outline RSD is a linear map identical to that of TAE ParaScope (see Appendix H.1), but this time mapping the normalized residual stream diffs to SONAR TAE embeddings, which contain the outline of the upcoming document.

We then evaluate this with an LLM-as-a-Judge (GPT-4o-mini, as before) and compare it against ground-truth outlines generated by another model following the same process, Gemma 3 27B Instruct (Gemma Team, 2025). We focus mainly on Coverage of Key Points, Ordering/Flow, Subject Match, Entities Match, and Details Match. See the full rubric in Appendix F.

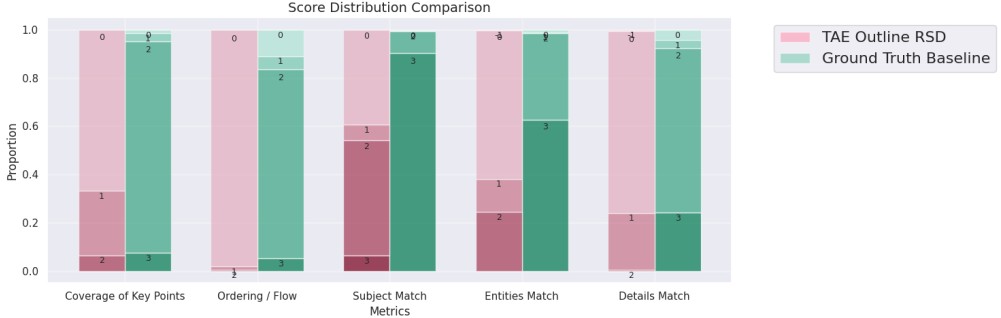

Figure 7: LLM-as-a-Judge comparison between outlines decoded by the TAE Outline RSD scored against the original Llama 3.2 70B outline. We compare against ground truth, which are the scores achieved by Gemma 3 27B with access to the full text & prompt.

We show results in Figure 7. We find that the TAE Outline RSD is able to decode some amount of information from the residuals stream, but that the performance is not as good as it was with the TAE ParaScope. We find that the subject match is generally high (61% at least minor match, 54% at least moderate match), but detail preservation is worse (24% minimal depth, i.e., very limited overlap), which is lower than for TAE ParaScope, which achieved minimal details 52% of the time.

Results on planning at the outline scale are relatively weak, aside from matching general subject matter. See example outputs in Appendix K. This gives some evidence for the Decodability Hypothesis, showing that the relatively small 3B-parameter model either does not plan at the outline scale or that such information is not linearly decodable. Experiments in the next section will provide some evidence for the former over the latter.

## 6 DEEPER ANALYSIS INTO INFORMATION AVAILABILITY

With this evidence for the Decoability Hypothesis, we apply the ParaScopes to understand when the model may be forming planning information.

## 6.1 Layer-wise Information Analysis

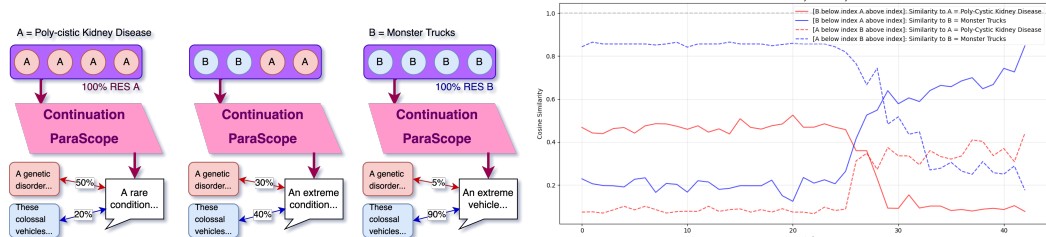

Figure 8: (left) explanation and (right) results of "layer scrubbing" for layer-wise information analysis

We employ a simple causal scrubbing methodology (Chan et al., 2022) with Continuation ParaScope to determine which layers contribute most to paragraph planning. See Figure 8 for illustration. We take the activations of the model across all the layers for two examples, and attempt to interpolate between the two examples to observe when the model chooses to write about one versus the other. In particular, we take the following prompt:

**Base Prompt:** "Tell me about cleaning your house in 50 words, then tell me about [A or B] in 50 words. Structure it as 2 paragraphs, 1 paragraph each."

We design our experiments using a controlled prompt structure where the model first writes about cleaning a house, then transitions to one of two dramatically different topics: either polycystic kidney disease (Paragraph A) or monster trucks (Paragraph B). This stark contrast allows us to distinguish which information is encoded at different layers.

For each layer $K$ between 0 and $N$, we perform a layer scrubbing procedure that isolates the contribution of different network depths from the paragraph transition token ("\n\n") . We use activations from $\text{RES}_A$ for layers 0 to $K$, and $\text{RES}_B$ for $K+1$ to $N$. See Appendix C.3 for details.

In Figure 8, this layer scrubbing analysis reveals a clear pattern of information processing. The early layers (0-25 out of 42 total) show minimal contribution to paragraph planning, with cosine similarity deltas below 0.05. The middle layers (25-35) demonstrate the strongest impact on future paragraph content, showing substantial cosine similarity deltas of 0.15 to 0.25. Finally, the output layer exhibits a distinct jump of approximately 0.1 in similarity delta, which we attribute to output embedding effects (directly influencing the first generation token) rather than planning computation. This suggests that layers 60%-80% into the model have the most planning-relevant activations.

## 6.2 Temporal Dynamics: When Planning Happens

To investigate when paragraph information appears, we analyze token-wise dynamics around paragraph transitions and attempt to understand when the model starts encoding follow-up information. We use Gemma-2-9B (Gemma Team, 2024) [1] and perform Continuation ParaScope on each of the different tokens, as shown in Figure 9.

We perform a few steps: 1) Generate paired paragraphs with controlled pairs of topics. 2) Extract residual streams from tokens within ±10 positions of "\n\n". 3) Apply Continuation ParaScope at each position. 4) Generate 64 possible completions per position. 5) Compare against original first or second paragraph using text embeddings. Results are shown in Figure 10

We first focus on an example where the initial topic is blockchain and the subsequent topic is ancient Mayan architecture, and in Table 6.2 compare the cosine similarities to both topics. We see that representation of the later topic only becomes significant at the newline token.

Averaging across 20 different pairs of topics, we then look at similarity to Topic 2. We find pre-transition it is 0.12 (±0.08), at transition it is 0.48 (±0.15), and post-transition it is 0.65 (±0.12). This gives moderate evidence that Gemma 2 9b is relatively just-in-time when it comes to forming plans in its activations, and mostly focuses on the specific immediate paragraph to write.

---

[1] we wanted to do a more fine-grained test of tokens "." and "\n\n", and llama 3 combines these into one token

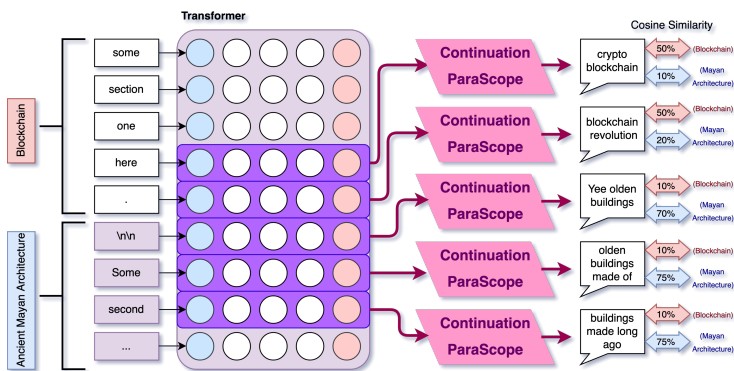

Figure 9: explanation of how we do token-wise analysis to see when the LLM prominently seems to be looking forward

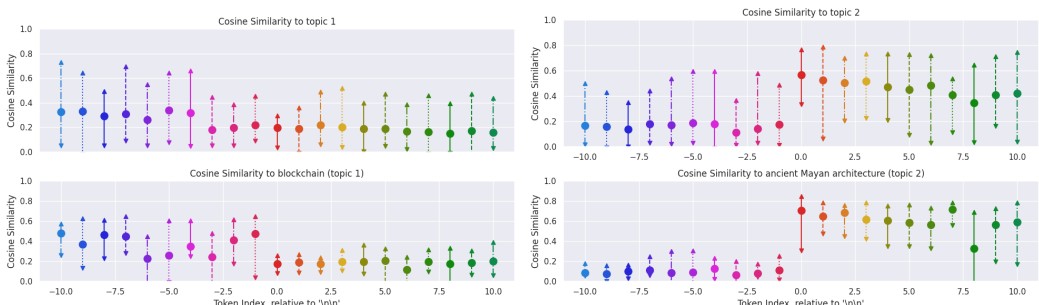

Figure 10: We get cosine similarity of Contination ParaScope generations at tokens near the newline token. We compare similarity to paragraph 1 (left) and paragraph 2 (right), for a specific example (bottom) and averaged over 20 examples (top)

|  | Pre-transition | Transition token | Post-transition |
|---|---|---|---|
| Topic 1 (blockchain) | 0.45–0.55 | 0.20–0.30 | 0.10–0.20 |
| Topic 2 (architecture) | 0.05–0.15 | 0.50–0.60 | 0.70–0.80 |

Table 1: Cosine similarity when using Continuation ParaScope at different positions relative to '\n\n'.

In Appendix G we present additional experiments to examine whether "\n\n" causes formation of plans in particular. We find evidence that it can in certain cases, but does not do so generally.

### 6.3 TESTING ON WIDER SET OF MODELS

We investigate the Decodability Hypothesis on a wider set of models and measure how paragraph-scale planning capacity changes with model size. We extend our experiments to the Gemma 3 (Gemma Team, 2025) family of models ranging from 270M to 27B in size. In Table 2, we find performance to be generally similar between model sizes, with TAE ParaScope seeming to perform better than Continuation ParaScope.

## 7 DISCUSSION

We have investigated how language models encode and utilize paragraph-scale prospective structure by probing their residual streams at transition points. Our paragraph-level Residual Stream Decoder methods, Continuation ParaScope and TAE ParaScope, reveal that models carry significant infor-

|        | Llama-3B | Gemma 270M | Gemma 1B | Gemma 4B | Gemma 12B | Gemma 27B |
|--------|----------|------------|----------|----------|-----------|-----------|
| TAE    | 0.534    | 0.484      | 0.485    | 0.501    | 0.498     | 0.512     |
| Cont.  | 0.390    | 0.307      | 0.390    | 0.373    | 0.331     | 0.341     |

Table 2: Cosine similarity (Standard error $\pm 0.002$) for ground truth, Llama-3B, and Gemma-3 models. These compare to Llama-3 baselines of Auto-decoded 0.922 and Regenerated 0.806

mation about upcoming content, comparable to having a limited "lookahead" of a 5-10 tokens, and find evidence for the Decodability Hypothesis. We additionally expand the method to a TAE Outline Residual Stream Decoder to probe for longer-context data, and found weak limited results, suggesting that while long-horizon prospective structure may be present, it is likely limited in Llama 3.2 3B.

More specifically, we found a linear TAE ParaScope model can often capture broad semantic signals from residual stream activations, such as topic or subject domain, while the Continuation ParaScope is more inconsistent, but offers greater textual coherence in the generated paragraphs and can sometimes reveal more specific details. Together, these findings shed light on the hidden mechanisms of paragraph-scale future-oriented information in large language models.

We additionally used Continuation ParaScope to investigate distribution of planning-related signals, and found it is not uniform across all layers. Instead, we see that the middle layers (roughly $60 - 80\%$ of the model's depth) concentrate these signals, in line with previous research (Meng et al., 2022). Earlier layers focus on local processing and later layers finalize immediate token decisions, whereas the middle layers integrate context to guide broader text generation.

Furthermore, when observing activations at different tokens, we find a sharp shift at paragraph boundaries: the model of study typically creates or refines its "plan" precisely when it processes the \n\n token. This myopic behavior suggests a form of just-in-time updating of anticipatory signals.

These results combine to start forming a picture of how future-content encoding behavior in LLMs works.

### 7.1 FUTURE WORK AND LIMITATIONS

We have so far focused on validation of various methods for Residual Stream Decoding, and found evidence for the Docodability Hypothesis. Since we use two different methods, and run additional tests with Continuation ParaScope, there is some evidence to rule out that the planning information we extract is completely spurious. However, there is room for further research into understanding the degree to which the planning signals we find are strictly causal to model performance.

Additionally, we have focused on short structured paragraphs and full-output outlines. These are not guaranteed to be the most natural scales when it comes to investigating models, and does not generalize cleanly to domains of math, code, and chemistry. It may be the case that there are other units of text, such as sentences, which are more "natural" to probe for. It may also be the case that planning information may be distributed across many tokens, and may not always be cleanly represented in one token.

Finally, our tests are primarily limited to Llama 3.2 3B and Gemma 3 models, and future work would investigate the degree to which planning differs in models of various different sizes beyond 27B.

Nevertheless, our results emphasize that large language models do maintain recognizable paragraph-level plans — albeit mostly confined to the immediate next section — and that these signals can be partially decoded. Future work may extend this approach to different horizons, investigate how these mechanisms interact with factual correctness or stylistic consistency, and refine interpretability techniques to further clarify the planning processes within large-scale transformer architectures.

These approaches may also lay the groundwork for tools such as monitoring of models in advance of writing an output, and may help in understanding a model's original intentions prior to completion of an output. We leave these applications to future work.

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

## A  DATASET GENERATION DETAILS

### A.1  NEXT-PARAGRAPH PREDICTION

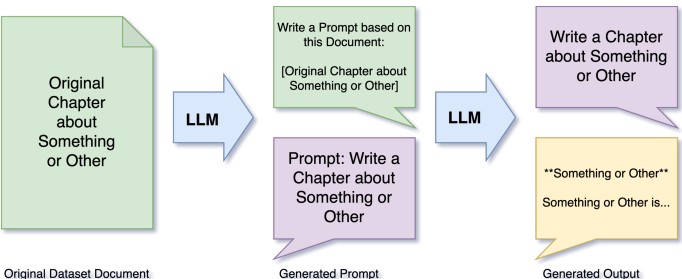

Figure 11: Multi-step dataset generation process.

First, we extract chunks from FineWeb Edu (Penedo et al., 2024), and use Gemma 2 27B (Gemma Team, 2024) to convert these into structured writing prompts of the form "Write a [type], titled [name], which includes [topics], approximately [length]". This intermediate step helps ensure diversity in our final dataset. We do this for 1 million text samples.

Here is the prompt used to generate the dataset of generated prompts:

```
Write a prompt based on the above text, that is a single-paragraph,
high-level description. Make the prompt in the format format similar to:
"Write a (news feed/chapter/piece/article/wiki entry/...),
titled (document name)', which includes (1-2 sentence list
of topics to cover, kept very vague). The full piece
should be approximately (n-paragraphs or other unit of length)".
```

We then take these prompts, and generate outputs with the model we are studying. Ie: Llama 3.2 3b instruct (Dubey et al., 2024). We split these by '\n\n' and the result is that we have 1 million model generations with approximately 10 million paragraphs which we use as our dataset of model paragraph outputs.

For Gemma 3 models of various sizes, we get 100,000 model generations for each model, outputting a total of around 900,000 paragraphs for each model.

For running evaluation, we use a holdout test set of 1000 prompts and generations that are not used during training.

### A.2 OUTLINE PREDICTION

For outlines, we take the model generations from before, and pass them to a new model. We tried various open source models for reproducability, including "openai/gpt-oss-120b", "meta-llama/Meta-Llama-3-8B-Instruct", "meta-llama/Llama-3.3-70B-Instruct", and "Qwen/Qwen2-72B-Instruct".

We found Llama 3 70B (Dubey et al., 2024) had the best results on the tradeoff between precision and brevity.

The prompt we used was as follows:

```
Return a short, high-level bullet-point outline of the main ideas from
the text you are given. Do NOT include any reasoning.

Rules:
- Make as 4-5 bullet points maximum
- Use numbers to enumerate the bullet points
- Aim to capture main ideas of the whole text in the bullet points
- At most 2 short subpoints per point
- Short phrases only (no lengthy sentences)
- Specific to this text (not generic).
```

## B BASELINES

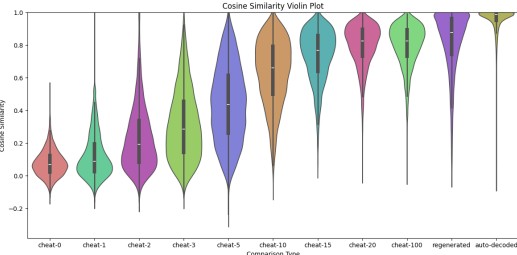

Figure 12: Comparison of how generations starting with some number of cheat tokens compare to regeneration and auto-encoding, using cosine similarity of text embed vectors.

Using the model generated dataset, we compare with the regenerated baseline with the cosine similarity of a text-encoding model (see Appendix C.1). We see that with 10 cheat tokens, the model is already very good at inferring what the model will say, and with 20 cheat tokens there is no substantial difference compared to adding even 100 cheat tokens.

There may be confounders. For example, it has been shown that text-embed models over index on the first few tokens of the embedded text (Lee et al., 2024; Wu et al., 2025). It is also possible that for very short texts, the model may choose to continue the paragraph when it would have counted the paragraph as completed were it to see the text in context. We leave further improvement of this methodology to future work.

## C EVALUATION

### C.1 AUTOMATED METRIC EVALUATION

We first evaluate our methods using established NLP metrics. Using the all-mpnet-base-v2 text embedding model, we compute cosine similarity between generated and reference texts. The SONAR ParaScope methods

(both Linear and MLP variants) achieve mean similarity scores of 0.51 (±0.20) and 0.50 (±0.20) respectively, significantly outperforming the Continuation ParaScope at 0.33 (±0.20). These results position both SONAR methods as comparable to the cheat-5 baseline (0.44), while falling short of full regeneration (0.82).

We supplement these findings with BLEURT scores, which show consistent patterns: SONAR ParaScopes achieve scores of 0.404 (±0.144) and 0.395 (±0.138), again comparable to the cheat-5 baseline at 0.396 (±0.102). The Continuation ParaScope achieves 0.364 (±0.135), significantly above random baseline (0.192 ±0.094) but below the regeneration baseline (0.619 ±0.216).

## C.2 Scoring Comparison with LLM-as-a-Judge

To complement the automated metrics, we performed a fine-grained evaluation using GPT-4o-mini as an evaluator[2], following the "LLM-as-a-judge" paradigm (Zheng et al., 2023a). We developed a detailed rubric covering multiple aspects of text quality and prompted the LLM to provide brief reasoning before assigning a score for each dimension, a technique shown to improve reliability in rubric-based LLM evaluation (Kim et al., 2024a; Gattani et al., 2024).

We choose 4 main aspects to focus on:

Key dimensions assessed included Coherence (evaluating flow and logical progression, scale 0-3), Subject Match (comparing topic similarity from unrelated to identical focus, scale -1 to 4), Entity Preservation (comparing specific entities mentioned, scale -1 to 4), and Detail Preservation (comparing the specificity of information, scale -1 to 3). The full rubric is provided in Appendix E.

This qualitative assessment revealed distinct trade-offs: the AutoEncoder Map ParaScope methods (Linear and MLP) demonstrated superior preservation of high-level semantic content (Subject and Entities), often matching or exceeding baselines like cheat-5/cheat-10 in topic relevance. Conversely, the Continuation ParaScope consistently generated more coherent and fluent text, scoring higher on the Coherence dimension, though it captured less specific subject matter and fewer entities compared to the reference paragraphs.

## C.3 Layer Scrubbing Details

For each layer $K$ between 0 and $N$, we perform a layer scrubbing procedure that systematically isolates the contribution of different network depths. We first extract residual stream activations $\text{RES}_A$ and $\text{RES}_B$ from the paragraph transition token ("\n\n") for both test conditions.

We then create hybrid activations by combining layers $[1...K]$ from one condition with layers $[K + 1...N]$ from the other, generating outputs in both directions to capture the full effect of layer-wise information transfer. For each configuration, we generate 100 samples and compare the resulting outputs against reference texts using 'all-mpnet-base-v2' (Song et al., 2020) embeddings.

To ensure our measurements reflect genuine information availability rather than limitations in our probing methodology, we generate 64 samples per condition and compare the best-matching outputs against references A and B.

# D   Additional Figures

## D.1   Filtered vs Unfiltered Comparisons of Scores

In addition to Figure 4, we plot graphs of the unfiltered data in Figure 13. That is, we allow examples that are "too short" for fair evaluation, such that the full examples are shown in cheat-10. Hence we see that as more tokens are exposed, cheat-5 and cheat-10 have some examples with perfect match.

# E   Full Rubric Details

In this section, we provide the complete evaluation rubric used to assess the quality and similarity of generated paragraphs in our experiments. This rubric was designed to capture multiple dimensions of text quality and semantic similarity between the reference paragraphs and those generated by our ParaScope methods.

## E.1   Complexity Assessment

- **0: Trivial** - Text contains minimal content (e.g., only section headers or placeholder text)

---

[2]Using 'gpt-4o-mini-2024-07-18' via the OpenAI API

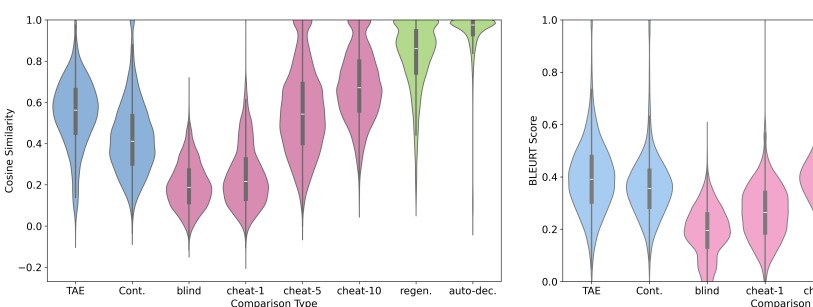

Figure 13: Violin plots showing the performance of TAE ParaScope and Continuation ParaScope against the baselines (0, 1, 5, and 10 cheat tokens) and ground truth (regenerated, auto-decoded) on Cosine sim (left) and BLEURT (right), including short examples < 20 tokens

- **1: Simple** - Basic content with minimal detail (e.g., simple section headers with brief descriptions)
- **2: Some detail** - Contains short, undetailed sentences about the topic
- **3: Many details** - Contains detailed paragraphs with specific information and nuanced content

## E.2 COHERENCE

- **0: Completely incoherent** - Text contains excessive repetition, nonsensical phrases, or strange symbols
- **1: Partially coherent** - Text is repetitive or has formatting issues (e.g., repeated key phrases, awkward pauses)
- **2: Mostly coherent** - Text has minor errors but maintains logical progression
- **3: Flawless flow** - Text demonstrates logical progression, clear transitions, and no repetition

## E.3 STRUCTURE

- **0: No alignment** - Structural mismatch (e.g., one is a title, the other a paragraph)
- **1: Partial overlap** - Some structural similarities but significant differences
- **2: Highly similar structure** - Matching structural elements and organization

## E.4 SUBJECT MATCH

- **-1: No subjects to compare** - Insufficient content for comparison
- **0: Completely unrelated subjects** - Topics from entirely different domains (e.g., "corporate law" vs. "particle physics")
- **1: Vaguely similar field** - Subjects from broadly related areas (e.g., "biology" vs. "physics" as sciences)
- **2: Related general domain** - Adjacent fields or related domains (e.g., "history" vs. "archaeology")
- **3: Same subject** - Both discuss the same general topic (e.g., "ancient Mayans")
- **4: Identical focus** - Both analyze the exact same specific topic (e.g., "ancient Mayan architecture")

## E.5 ENTITIES

- **-1: No entities to compare** - Insufficient entities for comparison
- **0: Completely unrelated** - Entities from different categories (e.g., "Norway" vs. "smartphone")
- **1: Vaguely similar category** - Entities of the same type (e.g., countries, people, cities)
- **2: Similar category** - Entities with categorical similarities (e.g., related countries, similar professions)
- **3: Partial identical entities** - Some matching entities with some differences
- **4: Almost all key entities match** - High degree of entity overlap between texts

### E.6 DETAILS

- **-1: Neither text has details to compare** - Insufficient details for comparison
- **0: Details differ completely** - No overlap in specific information provided
- **1: Minimal depth** - Basic shared details without specifics
- **2: Moderate depth** - Shared details with some supporting facts
- **3: Highly specific details** - Precise, quantitative, or technical details match

### E.7 TERMINOLOGY

- **-1: No terminology to compare** - Insufficient terminology for comparison
- **0: No shared terms** - Completely different vocabulary and terminology
- **1: Some overlap** - Partial matching of domain-specific terms
- **2: Domain-specific alignment** - Consistent use of field-appropriate terminology

### E.8 TONE

- **0: Mismatched** - Different registers or sentiment (e.g., clinical vs. casual, positive vs. negative)
- **1: Consistent** - Similar register, formality level, and sentiment

### E.9 IDENTICAL ASSESSMENT

- **0: Not identical** - Texts differ in content, even if similar
- **1: Identical** - Texts are essentially the same with only minor variations

This comprehensive rubric allowed us to systematically evaluate the quality of our ParaScope-generated paragraphs across multiple dimensions, providing a nuanced understanding of how well our methods capture and reproduce the planning signals present in the model's residual stream.

## F OUTLINE RUBRIC DETAILS

In this section, we provide the complete outline evaluation rubric used to assess the quality and similarity of generated outlines in our experiments. The rubric was designed to capture the key aspects of outline structure and semantic similarity between the reference outlines and those produced by decoding outline embeddings (SONAR). The embeddings are predictions resulting from a linear probe trained to map normalized residual stream diffs to SONAR TAE embeddings.

### F.1 COMPLEXITY ASSESSMENT

- **0: Trivial** - Text contains minimal content (e.g., only section headers or placeholder text)
- **1: Simple** - Basic content with minimal detail (e.g., simple section headers with brief descriptions)
- **2: Some detail** - Contains short, undetailed sentences about the topic
- **3: Many details** - Contains detailed paragraphs with specific information and nuanced content

### F.2 COHERENCE (OUTLINE-LEVEL)

- **0: Completely incoherent** - Excessive repetition, nonsensical phrases, or strange symbols
- **1: Partially coherent** - Repetitive or has formatting issues (e.g., repeated key phrases, awkward pauses)
- **2: Mostly coherent** - Minor grouping or ordering issues, but overall logical
- **3: Clear and coherent** - Logical outline structure with clarity

### F.3 HIERARCHY / STRUCTURE

- **0: No recognizable hierarchy** - Flat or malformed structure
- **1: Basic levels exist** - Often inconsistent
- **2: Mostly correct hierarchy** - Some mismatches present
- **3: Highly similar structure** - Matches closely with minimal deviations

## F.4 COVERAGE OF KEY SECTIONS

- **0: Most key sections missing** - Outline 2 omits or replaces core sections
- **1: About half present** - Roughly 50% of major sections covered
- **2: Most sections present** - Minor omissions or regroupings allowed
- **3: Full coverage** - All major sections appear (synonyms/regrouping acceptable)

## F.5 ORDERING / FLOW

- **0: Largely shuffled** - Illogical or inconsistent ordering
- **1: Partial overlap** - Frequent swaps in order
- **2: Mostly consistent** - Minor order swaps only
- **3: Closely aligned** - Order follows reference outline closely

## F.6 SUBJECT MATCH

- **-1: No subjects to compare** - Insufficient content for evaluation
- **0: Completely unrelated** - Topics from entirely different domains (e.g., "corporate law" vs. "particle physics")
- **1: Vaguely similar field** - Broad overlap only (e.g., "biology" vs. "physics")
- **2: Related general domain** - Adjacent or related fields (e.g., "history" vs. "archaeology")
- **3: Same subject** - Both discuss the same general topic (e.g., "ancient Mayans")
- **4: Identical focus** - Both focus on the exact same subject (e.g., "ancient Mayan architecture")

## F.7 ENTITIES / KEY CONCEPTS

- **-1: No entities to compare** - Insufficient key terms/entities
- **0: Completely unrelated** - Entities from unrelated categories
- **1: Same category, little overlap** - Entities belong to same type but differ
- **2: Partial overlap** - Some matching entities or synonyms
- **3: Mostly preserved** - Majority of key entities retained
- **4: Nearly identical** - Almost all entities preserved

## F.8 DETAILS

- **-1: No details to compare** - Neither outline provides details
- **0: Completely different** - Details differ entirely (e.g., benefits vs. generic notes)
- **1: Minimal depth** - Very limited overlap (e.g., one shared concept only)
- **2: Moderate depth** - Shared details plus some supporting facts
- **3: Highly specific details** - Quantitative or precise overlaps (e.g., percentages, statistics)

## F.9 CONCISENESS OF HEADINGS

- **0: Verbose** - Headings too wordy or sentence-like
- **1: Mixed clarity** - Some concise, others verbose
- **2: Mostly concise** - Headings generally outline-appropriate

## F.10 IDENTICAL ASSESSMENT

- **0: Not identical** - Outlines differ in content, even if similar
- **1: Identical** - Outlines essentially the same (ignoring trivial formatting)

| Original Outline Generate | TAE Outline Residual Stream Decoder |
|---|---|
| Outline: 1. Quantum Uncertainty Principle - Limits measurement precision 2. Implications of the principle - Challenges classical determinism 3. Quantum mechanics nature - Probabilistic and uncertain 4. Measurement impact - Collapses wave function 5. Fundamental concept | Summary: 1.Quantum involution principle - Uncertainty of the quantum work 2. Relativity - Uncertainty of the quantum work 3. Implications of boundedness - Theoretical estimation 4. Implications of the uncertainty of the quantum work 5. Implications of the quantum work 3. Implications of the uncertainty of the quantum work 4. Implications of the quantum work 5. |

Table 3: Comparison of Original Outline Generate vs TAE Outline Residual Stream Decoder - Example 5

## G  TEMPORAL DYNAMICS: TOKEN REPLACEMENT ANALYSIS

Given that the model does not seem to form plans for the next paragraph, we attempt to further understand what causes the model to plan for the next paragraph. We perform the same procedure as we did in Section 6.3 but when generating the residual stream, replace the input token with "\n\n" to see if the specific token is responsible for the model generating the plans.

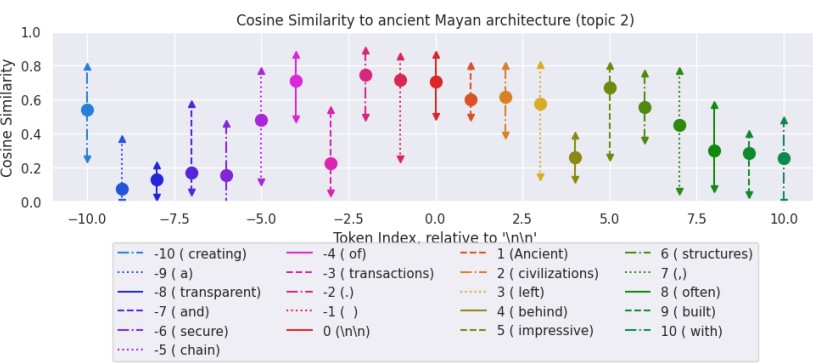

Figure 14: explanation of how we do token-wise analysis with manipulation to replace \n\n to see when the LLM prominently seems to be looking forward

We conduct detailed analysis of how the model handles artificially inserted paragraph transitions. For each text pair, we:

1) Take original text with natural "\n\n" transition 2) Replace tokens at positions [-10, +10] with "\n\n" 3) Apply Continuation ParaScope 4) Compare outputs with original second paragraph

Results show position-dependent effects: - Peak similarity at original transition (0.62 ±0.14) - Gradual decline pre-transition (0.45 ±0.12 at -5 tokens) - Steeper decline post-transition (0.31 ±0.13 at +5 tokens) - High variance in token-specific effects ($\sigma = 0.18$)

## H  SONAR LINEAR MAP

### H.1  SONAR PARASCOPE LINEAR MAP DETAILS

We train the model on a subset of 100,000 samples, taking the normalized residual stream activations as the input, and SONAR embeds of the paragraphs as the output.

Activations were normalized by using Welford's Online Algorithm on the first 10,000 texts (approx 100,000 samples) to compute mean and standard deviations, and using this to achieve approximately unit normal distribution on each dimension.

We train only on the output of the MLP and Attention (residual diff) of the final 12 layers (giving a total of 24 sub-layers) from Llama 3.2 3B's model's total of 28 layers.

In principle, one could use any text auto-encoder, or use a more complex probe such as an MLP probe.

Training hyperparameters were as follows:

```
- Linear map:
  - Batch size: 1,024
  - Weight decay: 1e-7
  - Learning rate: 2e-5 with x0.8 decay/epoch
  - Epochs: 10
```

## H.2 SONAR LINEAR MAP ANALYSIS

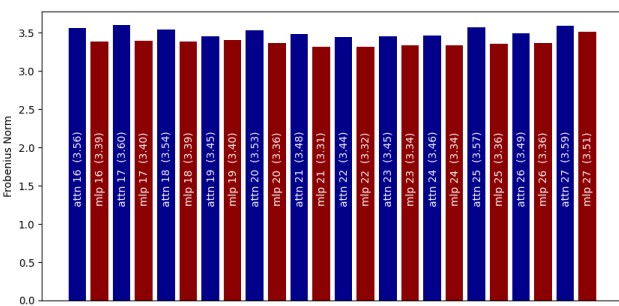

Figure 15: Layerwise Frobenius norm of Linear Regression Map from Sonar Map ParaScope

We perform detailed analysis of the learned SONAR mappings:

**Layer-wise Weight Distribution:** - Computed per-layer Frobenius norms - Attention layers: mean norm 3.56 (±0.42) - MLP layers: mean norm 3.39 (±0.38) - Layer-wise correlation $\tau = 0.73$

## H.3 CORRELATION ANALYSIS

We examine inter-method correlations using Kendall's $\tau$, including preliminary results with an MLP-based probe [3].

**Between Methods:** - MLP vs Linear: $\tau = 0.82$ - MLP vs Continuation: $\tau = 0.41$ - Linear vs Continuation: $\tau = 0.43$

**Between Metrics (Linear ParaScope):** - Length vs Subject: $\tau = 0.58$ - Coherence vs Detail: $\tau = 0.45$ - Entity vs Subject: $\tau = 0.61$

**Between Metrics (Continuation):** - Length vs Subject: $\tau = 0.39$ - Coherence vs Detail: $\tau = 0.31$ - Entity vs Subject: $\tau = 0.44$

These results suggest SONAR variants learn similar mappings but capture different aspects than Continuation ParaScope.

## I USAGE OF LLMS

We use LLMs at various parts of the process.

We use tools such as ChatGPT, Claude, Github Copilot, and Cursor when writing code and running experiments, and were iteratively used to create rubrics for scoring outputs, though all code is ultimately human-reviewed.

---

[3]This includes preliminary results where we trained an MLP Sonar ParaScope probe with 8192 hidden layers. We did not include MLP results these in the final paper as the results were not appreciably better than with the Linear model, and highly correlated with the Linear model.

For the literature review process of finding papers, tools like Google Scholar, Semantic Scholar, Connected Papers, we used, with supplemental search often done by ChatGPT, Claude, and Perplexity.

When it comes to paper writing, spell checker tools and LLMs were used to sometimes check for writing clarity and sometimes suggest edits that were accepted.

## J  EXAMPLES OF PARASCOPE OUTPUTS

We show a randomly selected set of 5 examples of TAE ParaScope and Continuation ParaScope examples, with the reference generation side by side.

| TAE | Cont. | Original |
|---|---|---|
| Understanding your migraine disorders is a key to managing and preventing migraine symptoms. By understanding your symptoms and preventing them, you can identify the best ways to manage and prevent migraine symptoms. By learning about your symptoms and preventing them, you can help manage your symptoms and prevent them. | Understanding the importance of sleep and establishing a consistent sleep schedule can have a significant impact on overall health and well-being. By making sleep a priority and creating a sleep-conducive environment, individuals can improve the quality of their sleep and wake up feeling refreshed and revitalized. Whether you're looking to improve your sleep hygiene or simply need some tips to help you get a good night's sleep, there are many resources available to help. So, take the first step towards better sleep and start making it a priority today. | Understanding your migraine triggers is the first step towards managing your symptoms and improving your overall quality of life. By exploring your individual triggers and taking action to manage them, you can reduce the frequency and severity of your migraines and take control of your health. |

Table 4: Comparison of TAE vs Cont. vs Original - Example 1

| TAE | Cont. | Original |
|---|---|---|
| The CDF International Conference on Poetry 2019 was a prestigious event held in Cologne, Germany, that brought together the world's leading scholars in poetry and literature. The event celebrated the 35th anniversary of the conference, which featured a comprehensive presentation of poetry and literature from across the globe. The conference was designed to bring together scholars from different backgrounds and disciplines. The conference featured a wide range of exhibitions on poetry, poetry, and literature. | The [Year] conference, held on [Date] at [Location], was a groundbreaking event that brought together leading experts in the field of [Field] to discuss the latest advancements and challenges in [Specific Area of Interest]. The conference featured a diverse range of speakers, including [Notable Speakers], who shared their insights and experiences on topics such as [Key Topics]. The event was well-attended by [Number] of delegates, who engaged in lively discussions and networking opportunities. | The 2015 CDI Poetry Conference was a premier event for poets and literature enthusiasts, held in the vibrant city of Toronto, Canada, from May 13th to 16th, 2015. The conference brought together over 200 poets, writers, and scholars from across Canada and around the world to share their work, engage in lively discussions, and explore the latest trends and themes in Canadian poetry. The conference featured an impressive lineup of notable attendees, including poet and Nobel laureate Margaret Atwood, critically acclaimed poet and essayist Tanya Talagant, and renowned publisher and literary critic Greg Gellenbeck. |

Table 5: Comparison of TAE vs Cont. vs Original - Example 2

| TAE | Cont. | Original |
|---|---|---|
| In the meantime, the new rules on investing are likely to lead to greater transparency and transparency in the decision-making process. | Overall, the recent developments in the field of quantum computing and the emergence of new technologies have created a new landscape for the development of quantum algorithms and applications. As researchers continue to push the boundaries of what is possible with quantum computing, we can expect to see new breakthroughs and innovations in the years to come. | In the meantime, startups and investors are already taking steps to prepare for the new rules. Many crowdfunding platforms are offering educational resources and training programs to help investors better understand the risks and benefits of investing in startups. |

Table 6: Comparison of TAE vs Cont. vs Original - Example 3

## K    EXAMPLES OF TAE OUTLINE RSD OUTPUTS

We show a randomly selected set of y5 examples of TAE Outline Residual Stream Decoder outputs.

| TAE | Cont. | Original |
|---|---|---|
| In the final analysis, the legacy of pluralism is a question of challenging and unique humanity. By combining the notions of progress and accountability, we are able to recognize the unique complexity of humanity. By pursuing the notions of responsibility and inclusiveness, we are able to foster a vision of the future. | In conclusion, the concept of "nothingness" is a complex and multifaceted idea that has been explored in various philosophical, scientific, and cultural contexts. From the perspective of physics, nothingness is often seen as the absence of matter and energy. However, from a philosophical standpoint, nothingness can be understood as a fundamental aspect of existence, a void that gives rise to the possibility of creation and meaning. Ultimately, the concept of nothingness challenges our understanding of reality and encourages us to think critically about the nature of existence. By exploring the concept of nothingness, we can gain a deeper understanding of the world and our place within | In conclusion, the concept of meritocracy is both a promise and a paradox. On the one hand, it has the power to inspire individuals to strive for excellence and promote social mobility. On the other hand, it can perpetuate existing power structures, reinforce competition, and overlook systemic inequalities. As we move forward in an increasingly complex and interconnected world, it is essential that we confront the paradox at the heart of meritocracy and begin to rethink our assumptions about success and achievement. Only by acknowledging the true nature of meritocracy can we create a more inclusive and equitable society that rewards achievement, rather than just talent. |

Table 7: Comparison of TAE vs Cont. vs Original - Example 4

| TAE | Cont. | Original |
|---|---|---|
| "We are looking forward to the opening of the new premises". "We are looking forward to the opening of the new premises. " | **Expected outcomes:** | "We're committed to providing our customers with a world-class storage experience," said [Name], President of Murphy Brothers Contracting. "Our partnership with BETCO Inc. has enabled us to push the boundaries of self-storage innovation, creating a facility that's not only functional but also environmentally friendly." |

Table 8: Comparison of TAE vs Cont. vs Original - Example 5

| Original Outline Generate | TAE Outline Residual Stream Decoder |
|---|---|
| Outline: | Summary: |
| 1. Rising college costs | - 1st Grade School Expenditure |
| - Increased student loan debt | - Increased Financial Liabilities |
| 2. Types of student loans | - 2nd Grade School Expenditure |
| - Federal and private loans | - Reasonable Income |
| 3. Consequences of student loan debt | - 3rd Grade School Liabilities |
| - Defaulting and credit damage | - Increased Expenditure |
| 4. Navigating student loan borrowing | - Privacy Policy |
| - Responsible borrowing practices | - 3rd Grade School Liability |
| 5. Prioritizing financial success | - Increased Expenditure |
| - Exploring financial aid options | - Increased Expenditure |
| | - Increased Expenditure |
| | - Increased Expenditure |
| | - Increased Expenditure |
| | - Increased Expenditure |
| | - Increased Expenditure |
| | - Increased Expenditure |
| | - Increased Expenditure |
| | - Increased Expenditure |
| | - Increased Expenditure |
| | - Increased Expenditure |
| | - Increased Expenditure |

Table 9: Comparison of Original Outline Generate vs TAE Outline Residual Stream Decoder - Example 1

| Original Outline Generate | TAE Outline Residual Stream Decoder |
|---|---|
| Outline: | Summary: |
| 1. Discovery of Homo floresiensis | 1.Discovery of small-scale dinosaurs |
| - 3 feet 7 inches tall | - Life-changing experiments at the University of Leeds |
| - Robust and resourceful species | |
| 2. Genetic analysis findings | 2.Classical humanities |
| - Interbred with other human species | - Explorations and discoveries of the 2nd-century dinosaur |
| - Hybrid offspring created | |
| 3. Implications for human evolution | - Common knowledge of anatomy |
| 4. Public lecture event | - Explorations and discoveries of the 3rd-century dinosaur |
| - Featuring Dr. Michael Morwood | |
| - Special exhibit on fossil remains | - Interesting connections to the living environment |
| 5. Event details | - Physiology of the 5th-century dinosaur |
| - Free and open to the public | - Explanation of scientific findings |
| - Registration recommended | - Participation of the population of the 5th-century dinosaur |
| | - Research Needs |
| | - Significance of natural discoveries and exploration |

Table 10: Comparison of Original Outline Generate vs TAE Outline Residual Stream Decoder - Example 2

| Original Outline Generate | TAE Outline Residual Stream Decoder |
|---|---|
| Outline: | Summary: |
| 1. Decline of labor conditions | 1. Labor market volatility |
| - Low wages and poor working conditions | - |
| 2. Shift to precarious labor market | 5. Employment inequality |
| - Rise of automation and globalization | - |
| 3. Negative future implications | 3. Disproportionate impact on labour market |
| - Widespread unemployment and inequality | - |
| 4. Need for policy change | 2. Ageing of labour and basic labour |
| - Support for workers in transition | - |
| 5. Transformation of economic systems | 4. Employment restrictions |
| - Prioritizing worker well-being | - |
| | 3. Disproportionate impact on productivity |
| | - |
| | 3. Changes in labour market conditions |
| | - |
| | 4. Necessity for innovative work |
| | - |
| | 5. Changes in the labour market |
| | - |
| | 5. Sustainability of human rights |
| | - |
| | 3. Necessity for improved labour market practices |

Table 11: Comparison of Original Outline Generate vs TAE Outline Residual Stream Decoder - Example 3

| Original Outline Generate | TAE Outline Residual Stream Decoder |
|---|---|
| Outline: | Summary: |
| 1. Pacific Flyway waterfowl trends | 1. Range of Pacific Waterfowl |
| - Mixed breeding conditions | - Decline in habitats |
| - Population increases and declines | 2. Range of Wild Waterfowl |
| 2. Regional population changes | - Increased Prevalence |
| - Western: Mallard increase, Canada Goose decline | 3. Bay and Swamp Areas |
| - Central: Wood Duck increase, Merganser decline | - Conservation Effects |
| 3. Eastern Pacific Flyway trends | 4. Pacific Waterfowl |
| - American Golden-Plover increase | - Increased Prevalence |
| - Harlequin Duck decline | 3. Habitats and Swamp Areas |
| 4. Alaska waterfowl trends | - Negative Flooding |
| - Snow Goose increase | 4. Environmental Effects |
| - Ross's Goose decline | - Conservation Effects on Birds |
| 5. Conservation importance | 5. Refugee Reservation |
| - Habitat protection | - Continued Changes |
| - Research and monitoring | 5. Requirements for monitoring and wildlife conservation |

Table 12: Comparison of Original Outline Generate vs TAE Outline Residual Stream Decoder - Example 4

| Original Outline Generate | TAE Outline Residual Stream Decoder |
|---|---|
| Outline:
1. Quantum Uncertainty Principle
- Limits measurement precision
2. Implications of the principle
- Challenges classical determinism
3. Quantum mechanics nature
- Probabilistic and uncertain
4. Measurement impact
- Collapses wave function
5. Fundamental concept | Summary:
1.Quantum involution principle
- Uncertainty of the quantum work
2. Relativity
- Uncertainty of the quantum work
3. Implications of boundedness
- Theoretical estimation
4. Implications of the uncertainty of the quantum work
5. Implications of the quantum work
3. Implications of the uncertainty of the quantum work
4. Implications of the quantum work
5. |

Table 13: Comparison of Original Outline Generate vs TAE Outline Residual Stream Decoder - Example 5

