# OpenReview forum: "ParaScopes: What do Language Models Activations Encode About Future Text?"
_ICLR.cc/2026/Conference — Submitted to ICLR 2026_

### Official Review · Reviewer_GVLL · 2025-10-20

**Soundness:** 2
**Presentation:** 3
**Contribution:** 2
**Rating:** 4
**Confidence:** 3

**Summary:**

The authors investigate the extent to which transformer residual stream activations at a new-paragraph token ("\n\n") contain enough information to reconstruct the transformer's next output paragraph (sampled autoregressively), which they refer to as "planning" for the next paragraph. They employ two different probes:
- "continuation ParaScope", which involves patching the residual stream activations at "\n\n" into a blank prompt and running the transformer forward autoregressively
- "TAE ParaScope", which involves fitting a linear mapping from residual stream activations to the SONAR embedding of the next paragraph.
The authors find that the performance of these probes beats random baselines and is roughly comparable to providing ~5 tokens of the ground-truth upcoming paragraph. They also extend these methods to document-outline-level information and find weaker results.

In addition, the authors study where planning takes place along the layer and position dimensions. They interpret their experimental results as showing that, for the model they study, 1) layers 60-80% into the model are most planning-relevant; 2) the model is relatively myopic and mostly focuses on the immediate next paragraph.

**Strengths:**

- The writing and especially the figures are clear. It is easy to understand what questions the paper investigates and its methodology for doing so.
- The performance of the two ParaScope probes are compared to various meaningful baselines.
- The layer-wise and temporal investigations in section 6 are interesting.

**Weaknesses:**

- In my view, this is a fairly incremental follow-up to an existing line of work. Previous works [1, 2] find that residual stream activations contain enough information to predict ~5 future tokens. Hence it is not so surprising that these activations can be used to reconstruct the next paragraph about as well as by providing ~5 ground truth tokens.
- The methods (activation patching, linear probes, text embeddings) are not especially novel, being straightforward adaptations of existing techniques to the specific next-paragraph setting.
- The notion of "planning" here is a fairly weak information-theoretic one, and it is possibly misleading to refer to this as "planning" at all. While the authors are careful to note this in the introduction, the writing elsewhere in the paper is sometimes more imprecise/anthropomorphic. E.g. 448: "myopic behavior suggests a form of just-in-time updating creation of plans";
- I'm not convinced that the results of the token-position experiments (6.2) show that the model is "relatively myopic". E.g. one cartoon picture consistent with the Continuation ParaScope results is that, before the "\n\n" token, the activations encode the plan "I will talk about blockchain until I reach a \n\n token, then switch to Mayan architecture."; then, after the token, simply "I will talk about Mayan architecture." This does not seem "myopic" to me.

[1] Pal et al. "Future Lens: Anticipating Subsequent Tokens from a Single Hidden State". 2023

[2] Cai et al. "Medusa: Simple LLM Inference Acceleration Framework with Multiple Decoding Heads". 2024

**Questions:**

- Are the results in Section 4 statistically significant? It's worth computing p-values for all the comparisons being made (e.g. "When looking at subject match, we find that TAE ParaScopes generally seem to outperform Continuation ParaScopes"; "when it comes to achieving at least Moderate Depth details [...], Continuation ParaScope (15%) is significantly better than the TAE ParaScope (3%)".)
- Error bars for Fig 8 (right) would also be appreciated.
- Any guesses as to how the qualitative differences between continuation and TAE ParaScope arise? Do they tell us anything about how models do planning?
- What is the significance of the 0.3 temperature? This introduces some irreducible noise into the ground truth next paragraph; is there a reason why 0 was not used instead?
- Can you reproduce the results of 6.1 and 6.2 using TAE instead of Continuation ParaScope?
- I would appreciate a bit more discussion on applications to monitoring and interpretability (though I realize this is mostly out-of-scope for this paper). When would ParaScope be more useful than simply monitoring the ground-truth next-paragraph output before it is used?

Nit:
- 030: Run-on sentence
- 044: malformed citation (Team)
- 068: malformed citation (team et al.)
- 121: "For our dataset, we require data that represents the outputs of the LLM so we can understand its thinking process." Somewhat confusingly and imprecisely phrased. You require data sampled autoregressively from the LLM because the ground truth is by definition the next paragraph sampled autoregressively.
- 162: "residuals diffs" -> "residual diffs"? I don't know what this term refers to. Are these the MLP/attention outputs?
- 185: K\in \{1, 5, 10\}
- 219: "ruberic" -> "rubric"
- 353: "generating outputs in both directions". Not clear what this means. I guess this means you tried both A below B and B below A?
- 560: malformed citation (Anonymous or Various)

---

> ### Author Response · Authors · 2025-12-03
>
> We are grateful to the reviewer for their time and for providing a clear summary of our work. We appreciate the detailed review and valuable feedback. We address the points below.
>
> ### W1 & W2:
> > This is a fairly incremental follow-up to an existing line of work.
>
> While it is true that our paper uses relatively simple methods, we think that this is more of a strength than a weakness. While one could possibly use more complex methods for trying to probe for planning in models, we think this could lead to confounding factors when analyzing models, and believe that the evaluation methodology for planning is clearly distinct to previous literature.
>
> ### W3:
> > The notion of "planning" here is a fairly weak information-theoretic one, and it is possibly misleading to refer to this as "planning" at all.
>
> Thank you for pointing this out, we have amended our work to have less references to 'planning' but believe that our work does capture an important aspect of the multi-faceted concept of planning. We do acknowledge that we still use "planning" in multiple places where needed.
>
> ### W4:
> > I'm not convinced that the results of the token-position experiments (6.2) show that the model is "relatively myopic".
>
> The experiments show that even when transferring the token activation directly before "\n\n", that the model outputs a newline token and continues to talk about topics more similar to blockchain, less similar to mayan architecture. It may be encoding that it should switch topic next paragraph, but (reasonably) doesn't seem to explicitly encode much local information about mayan architecture. (at least, our experiments show evidence against this). Perhaps "just-in-time" would be a better term to use rather than myopic, so we have updated the paper to reflect this.
>
> ### Q1:
> > Are the results in Section 4 statistically significant?
>
> Thank you for raising this. In response, we have added more information on standard errors throughout the paper.
>
> ### Small tweaks
> > 030: Run-on sentence - Thank you for providing feedback here, we have fixed this.
> > 044: malformed citation (Team), 068: malformed citation (team et al.) - Thank you. Fixed!
> > 219: "ruberic" -> "rubric" - We have corrected this, thank you.
> > 162: "residuals diffs" -> "residual diffs"? I don't know what this term refers to. Are these the MLP/attention outputs?
>
> Thank you for pointing this out,  we now use residual diffs throughout. For the Llama-3.2 models, we compute residual-mid diffs, meaning the change in the residual stream after each layer’s attention block.
>
> For the Gemma family, we use residual-pre diffs, i.e., the difference between consecutive pre-layer residuals, which captures the total change by the full transformer layer.
>
> In practice, we found that both approaches perform similarly, with the pre-diff approach being slightly simpler.

---

### Official Review · Reviewer_WGtw · 2025-10-30

**Soundness:** 2
**Presentation:** 2
**Contribution:** 2
**Rating:** 4
**Confidence:** 3

**Summary:**

This paper introduces ParaScopes, a framework designed to investigate the extent to which language models encode information about future text within their internal activations. The authors propose the Planning Decodability Hypothesis, which operationalizes the concept of model planning into a testable claim: that information about upcoming content is decodable from activations before that content is generated. The study introduces two Residual Stream Decoder methods, Continuation ParaScope and TAE ParaScope, to probe for paragraph-level and outline-level plans. Experiments on a Llama 3.2 3B model show that limited information, comparable to a 5-10 token lookahead, can be decoded, with planning signals being most prominent in the model's middle layers and appearing "just-in-time" at paragraph transition tokens.

**Strengths:**

- The paper introduces two distinct and complementary probing methods, the intervention-based Continuation ParaScope and the mapping-based TAE ParaScope, which strengthens the validity of its conclusions.
- The study goes beyond simply finding evidence of planning by localizing the relevant signals, identifying the middle layers of the network as the primary location for paragraph-level planning information.
- The paper provides a specific temporal account of planning, presenting evidence for a "just-in-time" mechanism where plans are sharply formed or refined at the moment of paragraph transitions.

**Weaknesses:**

- The findings are based almost entirely on a single, relatively small (3B parameter) model, making it unclear if they apply to larger, more capable architectures.
- The main TAE ParaScope method uses a linear map, which may be too simple to extract more complex, non-linearly encoded plans from the model's activations.
- The work primarily establishes that future-looking information is present in activations, but provides limited evidence that this information is causally used by the model to guide generation.

**Questions:**

1. Is the observed just-in-time planning an artifact of the 3B model's scale, or do you expect it to persist in larger models?
2. Have you tried a steering experiment by injecting a synthetic plan embedding to test for causality?
3. What is your hypothesis for why the Continuation ParaScope is sometimes better at fine-grained details than the TAE ParaScope?

---

> ### Author Response · Authors · 2025-12-03
>
> We thank the reviewer for their thoughtful and constructive feedback.
>
> We address their points raised below.
>
> ### W1:
> > The findings are based almost entirely on a single, relatively small (3B parameter) model, making it unclear if they apply to larger, more capable architectures.
>
> Thank you for bringing this up. We agree that evaluating additional models of different sizes would strengthen the empirical support for our findings. Accordingly, we have now run the Gemma family models (270 M–27 B) and included the new results in the latest revision of the paper, section 6.3. Interestingly and contrary to our expectations, we do not see a consistent increase in “planning signal” (as measured by our TAE ParaScope) with larger model size. Instead, we mostly find that results are quite similar for all models. For some larger models (for example, Gemma 27B compared to Gemma 1B) the probe performs slightly worse than it does on smaller models, but the differences are small.
>
>
> ### W2:
> > The TAE ParaScope method uses a linear map, which may be too restrictive to capture complex patterns.
>
> Thank you for raising this. We acknowledge that using a linear map for our TAE ParaScope could be seen as too simplistic to recover non-linearly encoded planning information. However, we intentionally chose a linear probe to avoid confounds associated with more complex probes. A non-linear probe might simply learn multi-step correlations, conflating what’s encoded in the model with what the probe itself figures out. We agree that going beyond a linear probe may be interesting for future work.
>
> ### Q1:
> > Is the observed just-in-time planning an artifact of the 3B model's scale, or do you expect it to persist in larger models?
>
> Thank you for raising this question. We have added some experimentation on larger models in a new Section 6.3, and generally found most results remain quite similar between different Gemma models of sizes 270 million to 27 billion parameters, so would mostly expect this to be the case. We expect that these experiments would inspire much similar research into model behaviour across model sizes and text timescales.
>
> Theoretically though, we mostly expect that since the model at any point is doing next-token prediction, adding too much information about a next paragraph when not needed to predict things in the current paragraph could degrade immediate performance, so it would make sense that the just-in-time behaviour would persist.
>
> ### Q2:
> > Have you tried a steering experiment by injecting a synthetic plan embedding to test for causality?
>
> Thank you for the suggestion of a steering experiment via synthetic plan-embedding injection to test for causality. We fully agree this would be a very interesting and useful experiment, and an example of how our methodology could be expanded upon. Unfortunately, this is somewhat out of scope for the immediate paper, but we think it would be interesting to explore it in follow-up research.

---

### Official Review · Reviewer_CuEw · 2025-10-31

**Soundness:** 1
**Presentation:** 2
**Contribution:** 1
**Rating:** 0
**Confidence:** 3

**Summary:**

This paper proposes two new methods for probing the "forward looking" or "planning" information present in language model hidden states. The common setup in the paper is given some text, bisect it, feed the first half through a target model you wish to study, take the hidden states of the last token in the first half and then try to generate the second half of the text from just those hidden states. The first method they propose, Continuation ParaScope, uses the target model to generate the second half of the text. The second method they propose, TAE Parascope, first trains a TAE on hidden states and generations from the target model and then uses the trained TAE to generate the second half of the text.

**Strengths:**

* The layer-wise and temporal dynamics studies in sections 6.1 and 6.2 provide some interesting analysis of language model hidden states
* Method diagram are a helpful aid to the writing

**Weaknesses:**

* Experimental setup for the main experimental results (in Figure 4) is problematic. They use data generated by the target model to then evaluate the target model. Obviously text sampled from the target model is likely under the target model. This introduces a major source of confounding. I expect a language model would generate the same or similar next couple tokens only given its last hidden state. Not necessarily because it has plan them out but possibly because its just the likely next thing to say given what is captured in that current hidden state. Given this it is unclear to me that the experiment actual measures the amount of "planning" information present in the hidden state.
* There is a second issue in the main experimental results (in Figure 4). In the description of the Cheat-K baseline they state "For fairness, we filter examples where more than 50% of the text would be exposed" this is actually completely unfair. For a fair comparison all methods should be evaluated on the same evaluation set. It is improper to remove these examples at all in my opinion, the whole point of that baseline is it cheats, but at the very least if you are going to remove them you need to remove the examples for all methods. You cannot compare two methods if they are evaluated on different subsets of the data.
* The results displayed in the violin plot would be much more convincing if demonstrated to be significant with a statistically test.
* The paper presents parascope as a general method for probing information, but only provides experiments with one model.
* There are a number of figures which are never referenced in the text. If the figure is important enough to be in the main paper it should be discussed and referenced.
* The presentation also could be improved. The text is redundant in places (e.g. splitting the paragraphs by "\n\n" is described on lines 035-036, 113-115, 125-128). The text in figures is often very small, and sometimes overlaps itself (e.g. Figure 4, Figure 5, Figure 7, Figure 8, Figure 10). There is a lot of whitespace around some diagrams (e.g. Figure 1, Figure 8).
* Some very minor details. The citation (team et al., 2024) on line 068 I believe should be capitalized "Team", the white grid lines in Figure 5 make the plot harder for me to read (I think because the bars are translucent), and Figure 10 x axis is labeled Token Index but the values on the x axis are decimal values 2.5, 7.5, etc.

**Questions:**

* What is the training/testing setup for the TAE Parascope? Where does the training data come from? Assuming the dataset constructed for evaluating Parascope variants is held out for unbiased evaluation.

---

> ### Author Response · Authors · 2025-12-03
>
> We thank the reviewer for taking the time to read our work and for the summary of our proposed methods (Continuation ParaScope and TAE ParaScope) as well as valuable feedback for improvement. We appreciate the recognition of the conceptual setup and happy to hear you found the diagrams helpful to understand our paper's setup.
>
> We address the points raised below.
>
> ### W1
> > Experimental setup for the main experimental results (in Figure 4) is problematic.
>
> Thank you for raising your concern regarding evaluation in Figure 4. Our experiment's aim is to test a more conservative question "given the model’s own internal trajectory, does its hidden state contain information about future text beyond what would be predicted purely from local next-token likelihoods". To reduce the confounding we (i) use independent probes, not the model’s own logits; (ii) evaluate predictions of future (not next) text spans.
>
> We acknowledge that model-generated data may still carry residual biases and understand that our current experiments are not sufficient to rule things out either way. However, we disagree with the comment in general, and believe that our experiments do make significant methodological progress on evaluating planning in LLMs, and do provide some evidence of some form of planning in LLMs.
>
> ### W2:
> > There is a second issue in the main experimental results (in Figure 4).The description of the Cheat-K baseline.
>
>
> Thank you for your suggestion here. We have tried to improve the clarity now by using the same filtering for all data in the main plot, and by including a separate graph in the appendix with all the unfiltered data points that are excluded for being too short. While this did not qualitatively change the results much, we do agree this is an unnecessary confusion.
>
> ### W4:
>
> > The paper presents Parascope as a general method for probing information, but only provides experiments with one model.
>
> Thank you for the feedback. We agree that evaluating additional and different size models would strengthen our empirical support. Accordingly, we ran the Sonar Parascopes experiments on the Gemma family of models (270M–27B) and reported the results in the most recent paper revision, section 6.3.
>
> ### Q1:
> > There are a number of figures which are never referenced in the text — could you clarify their role?
>
> We have now added references to all the figures in the text.
>
> ### Q2:
> > The presentation also could be improved. The text is redundant in places, The text in figures is often very small, and sometimes overlaps itself.
>
> Thank you for pointing out the above issues to us, we have removed the redundant explanations of the paragraph splitting method!
>
> ### Q3:
> > What is the training/testing setup for the TAE model? Where does the training data come from?
>
> We train on a dataset of same-language-model responses as described in Appendix A. We have a holdout data subset of 1000 prompts for evaluation, and test all the methods and baselines with this held-out data.
>
> ### Corrections:
> > Some very minor details.
> > The citation (team et al., 2024) on line 068 I believe should be capitalized "Team", the white grid lines in Figure 5 make the plot harder for me to read (I think because the bars are translucent), and Figure 10 x axis is labeled Token Index but the values on the x axis are decimal values 2.5, 7.5, etc..
>
> We appreciate your suggestions and the opportunity to improve the clarity of the writing and the visibility of the plots. We have corrected the “Team” citations now referencing the correct teams e.g. Anthropic Interpretability Team, Gemma Team etc.

---

### Official Review · Reviewer_Q1Md · 2025-11-01

**Soundness:** 3
**Presentation:** 3
**Contribution:** 2
**Rating:** 4
**Confidence:** 4

**Summary:**

The paper investigates whether large language models exhibit evidence of planning within their internal representations. The authors operationalize planning as the presence of information about future tokens within the model's residual stream. To test this “Decodability Hypothesis,” they introduce two decoding methods, termed ParaScopes, which attempt to reconstruct upcoming paragraphs from residual stream activations. By comparing these decoders against several baselines, the study finds some but limited evidence that model activations contain information about upcoming paragraph.

**Strengths:**

- The paper is well-written with a clear structure. Both the methodology and experimental setup are presented in a straightforward, easy-to-follow manner.
- The authors offer a valuable methodological advance by operationalizing planning as the decodability of upcoming text from residual stream activations.

**Weaknesses:**

- The paper equates the ability to decode future tokens or paragraphs from residual stream activations with evidence of "planning." This interpretation is not convincing to me. The residual stream contains contextual information that makes certain continuations more likely, even if no explicit future information is stored. For example, if the context describes the first half of a soccer match, the model predicting the second half next is just a result of coherence, not evidence of a stored "plan".
- The improvements over random or baseline methods are marginal. The results suggest shallow correlations rather than robustly decodable "plans."
- The core experiments are performed on a single mode, Llama 3.2 3B. Expanding to other family models, and especially to larger models, would be valuable.
- The "subject match" in outline-level experiments is insufficient to claim planning, and the other measures perform poorly.

**Questions:**

Why do you use temperature>0? By doing this, each continuation you sample, both in the dataset as for Continuation ParaScope can lead to different paragraphs. Would it make sense to resample each continuation multiple times and consider that when computing the metrics? Or using temperature=0 instead?

Your operationalization defines planning as decodable information about future text within the residual stream. How do you distinguish this from contextual predictability, i.e., the residual stream encoding contextual information that makes certain future tokens plausible rather than future planning?

Comments:
- line 219: ruberic-based -> rubric-based
- line 308 "we show the results in 6." -> "we show the results in Figure 7"

---

> ### Author Response · Authors · 2025-12-03
>
> We want to thank you for taking the time to review our paper and for providing constructive feedback. We are pleased to hear that you found our work clearly written and well-structured. We’re glad that the methodological contribution is useful and that our evaluation of planning capabilities resonates with you. Thank you for this recognition.
>
> We address your points below.
>
> ### W1 and Q2:
> > “The paper equates the ability to decode future text with the ability to plan…”
> > Your operationalization defines planning as decodable information about future text within the residual stream.  How do you distinguish this from contextual predictability, i.e., the residual stream encoding contextual information that makes certain future tokens plausible rather than future planning?
>
> We agree this is a concern and briefly touch upon this in the introduction. It is true there are likely degrees to which planning could be construed as "explicit" and purely correlational or “implicit”. Some previous work (Wu et al 2024) has shown some evidence of both being true to some degree. We try to limit the degree to which implicit correlational information is important by using very simple methods (linear map TAE parascope, constant activation transfer for continuation parascope) that would not be able to do complex inference for this reason.
>
> ### W2:
> > The improvements over random or baseline methods are marginal.
>
> We apologize for the lack of clarity here. We do show that the methods are significantly better than random, and provide various baselines up to and including a whole text ground truth, so it would make sense that the methods are not an improvement compared to a ground truth. See Appendix B for more information about this.
>
> ### W3:
>
> > The core experiments are performed on a single dataset and domain.
>
> Thank you for the feedback. We agree that evaluating a broader set of models, including larger ones, strengthens the empirical support for our claims. In response, we have run the Sonar Parascopes experiments on the Gemma family, spanning models from Gemma 270M up to Gemma 27B. We include the resulting findings in the updated version of the paper, section 6.3.
>
> ### W4:
>
> > The “subject match” in outline-level experiments is not well justified.
>
> We agree that the outline-level experiments are weaker, as written in the original paper: "Results on planning at the outline scale are relatively weak, aside from matching general subject matter." We see these experimental results as aspirational, and provide the novel evaluation methodology to inspire future work.
>
> ### Q1:
>
> > Why do you use temperature > 0?
>
> In order to have a fair baseline for the model, we compare the paragraph to a regenerated paragraph. To make conditions fair and identical, we use the same model, context, and temperature, and for this to be meaningful, we need the temperature to be non-zero. Any variance in different continuations should then be captured by the metrics we get for regeneration, so doing multiple generations per sample is not needed.
>
> ### Corrections:
>
> Thank you for pointing out the above issues to us, we have adjusted the writing!
>
> #### References
> Wilson Wu, John X Morris, and Lionel Levine. Do language models plan ahead for future tokens? arXiv preprint arXiv:2404.00859, 2024.

---

### Meta-Review · Area_Chair_jo3c · 2026-01-05

**Summary:**

The authors propose and evaluate probing token activations for information about future text (subsequent tokens). They offer their mostly positive results about the ability to predict such information as evidence for the "decodability hypothesis" they advance, which states that "planning" exists at scale X (e.g., paragraph) if a probe can recover information about content at this scale (e.g., the next paragraph) before it is generated.

Reviewers agreed that this paper was easy to follow and delineates an explicit operationalization of "planning" (though there was some objection to this framing, from Q1Md in particular).

However, the treatment of "planning" here needs further clarification, given that the authors have made this the core of their contribution. And all reviewers noted that the presented experiments focussed on a single model (this issue addressed by the authors in rebuttal).  Further, the approach is incremental given prior related work (per GVLL). The work as is may have limited impact as a result of these limitations.

**Reviewer Concerns:**

The authors extended their analysis to other models; this was a limitation raised by all reviewers.

**Reviewer Scores:**

Hard to say, but it's conceivable that WGtw in particular may have raised their score given that the single model was their main concern and this was addressed.

---

### Decision · Program_Chairs · 2026-01-26

Reject